# Oxygen Vacancy-Enhanced Ni_3_N-CeO_2_/NF Nanoparticle Catalysts for Efficient and Stable Electrolytic Water Splitting

**DOI:** 10.3390/nano14110935

**Published:** 2024-05-26

**Authors:** Xianghao Meng, Xin Zhao, Yulin Min, Qiaoxia Li, Qunjie Xu

**Affiliations:** 1Shanghai Key Laboratory of Materials Protection and Advanced Materials in Electric Power, College of Environmental and Chemical Engineering, Shanghai University of Electric Power, Shanghai 200090, China; 13685668047@163.com (X.M.); zx15383416084zx@163.com (X.Z.); minyulin@shiep.edu.cn (Y.M.); 2Shanghai Institute of Pollution Control and Ecological Security, Shanghai 200090, China

**Keywords:** electrocatalyst, hydrogen evolution reaction, oxygen evolution reaction, oxygen vacancies, water splitting

## Abstract

Highly efficient and cost-effective electrocatalysts are of critical significance in the domain of water electrolysis. In this study, a Ni_3_N-CeO_2_/NF heterostructure is synthesized through a facile hydrothermal technique followed by a subsequent nitridation process. This catalyst is endowed with an abundance of oxygen vacancies, thereby conferring a richer array of active sites. Therefore, the catalyst demonstrates a markedly low overpotential of 350 mV for the Oxygen Evolution Reaction (OER) at 50 mA cm^−2^ and a low overpotential of 42 mV for the Hydrogen Evolution Reaction (HER) at 10 mA cm^−2^. Serving as a dual-function electrode, this electrocatalyst is employed in overall water splitting in alkaline environments, demonstrating impressive efficiency at a cell voltage of 1.52 V of 10 mA cm^−2^. The in situ Raman spectroscopic analysis demonstrates that cerium dioxide (CeO_2_) facilitates the rapid reconfiguration of oxygen vacancy-enriched nickel oxyhydroxide (NiOOH), thereby enhancing the OER performance. This investigation elucidates the catalytic role of CeO_2_ in augmenting the OER efficiency of nickel nitride (Ni_3_N) for water electrolysis, offering valuable insights for the design of high-performance bifunctional catalysts tailored for water splitting applications.

## 1. Introduction

Over the past few decades, global energy consumption has increased at an alarming rate. The depletion of fossil fuels is driving our world toward a severe energy shortage. Consequently, the quest for novel energy sources holds immense importance for humanity’s sustainable growth. Hydrogen is the most ideal of all renewable energy sources because it has zero pollution and high energy density [1]. Among the hydrogen-forming methods, electrolytic water has attracted much attention from researchers as a result of its high utilization rate and lack of by-products. The electrochemical process of water splitting encompasses two distinct half reactions: HER at the cathode and OER at the anode. Precious metal catalysts (platinum, palladium, ruthenium, iridium) are active in hydrogen and oxygen evolution reactions but are currently expensive and difficult to apply in the industry [2,3]. Therefore, researchers must develop efficient and inexpensive electrocatalysts for electrochemical water separation [4].

When the element N is integrated into transition metal lattices, transition metal nitrides (TMNs), a type of mesenchymal substance, emerge, sharing characteristics with covalent compounds, ionic crystals, and transition metals [5]. Nanostructured TMNs, characterized by a large surface area, can be synthesized via temperature-regulated nitriding of oxide precursors by means of a “local structured reaction”, ensuring the oxide precursor’s crystal structure remains intact under stringent nitriding scenarios [6]. The introduction of N leads to an expansion of the metal lattice, a rise in metal spacing, and a reduction in the interaction force among metal atoms, culminating in the d-band shrinking and a rearrangement of state density near the Femi level [7,8]. Consequently, techniques like defect engineering, alloying, heteroatom doping, and the heterostructuring of TMNs represent sophisticated and beneficial approaches for nitride-based electrocatalysts [9]. As the count of valence electrons rises, the structure undergoes corresponding alterations [10]. TMNs are characterized by their distinct electronic configurations, featuring superior electrical conductivity, chemical robustness, electrocatalytic properties, and mechanical steadiness, making them versatile for various applications [11,12]. Typically, TMNs are characterized by their covalent, ionic, and metallic bonds, particularly those between transition metals and nitrogen atoms, which enlarge the parent metal lattice and reduce the size of the metallic d-band. The incorporation of nitrogen atoms in TMNs enhances their corrosion resistance, enabling them to function effectively in both acidic and alkaline environments. Compared to other compounds containing P, S, and Se, TMNs have advantages such as a large specific surface area, high chemical stability, and good electrical conductivity [13]. Ni_3_N, as a representative transition metal nitride (TMN), has been widely recognized as an electrocatalyst owing to its properties akin to those of noble metals, demonstrating respectable performance in the OER and HER [14,15,16]. Nevertheless, the intricate reaction mechanisms and high dissociation energy of water restrict its effectiveness. Presently, a range of augmentation techniques, including the formation of heterostructures, doping with heteroatoms, defect engineering, and alloying, have been employed to elevate the catalytic efficacy of Ni_3_N through the optimization of adsorption energies of intermediates. Despite these efforts, the catalytic performance of Ni_3_N remains significantly inferior to that of noble metal benchmarks in water splitting applications [17]. Concurrently, cerium dioxide (CeO_2_), recognized for its copious oxygen vacancies and the flexible valency shifts between Ce^3+^ and Ce^4+^, serves as an efficacious co-catalyst, augmenting both HER and OER efficiencies. Additionally, CeO_2_ is capable of activating water molecules in alkaline environments, boosting HER performance. Forming a composite structure is indeed helpful for electrocatalysis. Heterostructures have the following advantages in the field of electrocatalysis: enhancing catalytic activity, promoting charge separation, optimizing band structure, regulating interface properties, and improving stability [18]. Considering the unique benefits of Ni_3_N and CeO_2_, amalgamating them into a heterostructure might offer a viable strategy for the development of high-efficiency bifunctional catalysts [19,20].

Herein, we produced Ni_3_N-CeO_2_/NF heterostructure catalysts. Due to the synergy between Ni_3_N and CeO_2_, Ni_3_N-CeO_2_/NF demonstrates superior catalytic proficiency in the context of water electrolysis, achieving remarkably low overpotentials of merely 42 mV at 10 mA cm^−2^ for the HER. Particularly, for the OER, the material exhibits an ultra-low overpotential (η_50_ = 350 mV). Through EPR testing, we can see that the catalyst also has abundant oxygen vacancies. The in situ Raman spectroscopy indicates that cerium dioxide (CeO_2_) significantly facilitates the formation of nickel oxyhydroxide (NiOOH) enriched with oxygen vacancies (O_Vs_) during the OER. This comprehensive elucidation of the catalytic mechanism offers crucial insights for the subsequent development of heterostructured catalysts [21,22].

## 2. Material and Methods

### 2.1. Synthesis of Composites

Synthesis of Ni(OH)_2_-CeO_2_/NF Precursors on Ni foam. First, clean nickel foam was prepared, which was ultrasonically treated with 2 M HCl for 20 min to eliminate the oxides, and then ultrasonically treated with DI water and ethanol for 20 min. Then, we placed the clean nickel foam (2.0 cm × 4.0 cm) in a 50 mL Teflon autoclave and added 0.9 mmol Ni(NO_3_)_2_·6H_2_O, 1.8 mmol Ce(NO_3_)_3_·6H_2_O, 6 mmol CH_4_N_2_O, 4 mmol NH_4_F, and 30 mL DI water; after that, the mixture was stirred for 30 min to fully dissolve the medicine. Then, the Teflon autoclave was placed in an electric oven and maintained at 120 °C for 6 h. After dropping to room temperature, the precursor was rinsed with DI water several times and then placed in a vacuum drying oven at 70 °C to dry for 12 h.

Synthesis of Ni_3_N-CeO_2_/NF. A combustion boat containing Ni(OH)_2_-CeO_2_/NF precursors was calcined at 350 °C under ammonia atmosphere for 3 h in a tube furnace, and Ni_3_N-CeO_2_/NF was then obtained. Because the molar ratio of the synthesized Ce and Ni is different, we name it Ni_3_N-xCeO_2_/NF (Figure 1).

Synthesis of Ni_3_N/NF. First, we placed the clean nickel foam (2.0 cm × 4.0 cm) in a 50 mL Teflon autoclave and added 0.9 mmol Ni(NO_3_)_2_·6H_2_O, 6 mmol CH_4_N_2_O, 4 mmol NH_4_F, and 30 mL DI water; after that, the mixture was stirred for 20 min to fully dissolve the medicine. Then, the Teflon autoclave was placed in an electric oven and maintained at 120 °C for 6 h. After dropping to room temperature, the precursor was rinsed with DI water several times and then placed in a vacuum drying oven at 70 °C to dry for 12 h. Then, the precursor was heated to 350 °C for 3 h under ammonia atmosphere.

Synthesis of CeO_2_/NF. The synthesis process is the same as for Ni_3_N-CeO_2_/NF except that Ni(NO_3_)_2_·6H_2_O is not used.

### 2.2. Electrochemical Measurements

The electrocatalysts, synthesized and supported on nickel foam, were directly implemented as the working electrodes. The mercury/mercuric oxide electrode served as the reference electrode, while a polished graphite rod functioned as the counter electrode. The exposed area of the working electrode in contact with the electrolyte measured 1 cm^2^. Linear sweep voltammetry (LSV) measurements were conducted with a scan rate of 5 mV s^−1^. The obtained polarization curves were adjusted for iR compensation to account for the ohmic resistance presented by the electrolyte, with a compensation level set to 100%. Cyclic voltammetry (CV) profiles were acquired by executing CV assays across a spectrum of scan rates (20, 40, 60, 80, and 100 mV·s^−1^) within the potential windows from 0.1 to 0.2 V (vs. RHE) for the HER, and from 1.06 to 1.16 V (vs. RHE) for the OER. The stability performance test was conducted at 50 mA cm^−2^ for 40 h (not iR-corrected). Furthermore, the evaluation of overall water electrolysis was carried out utilizing a two-electrode configuration, employing two identical catalyst-coated electrodes serving concurrently as both the anode and cathode. The stability of the system was assessed at 10 mA cm^−2^ over a continuous duration of 34 h, with the measurements taken without applying iR compensation. Electrochemical impedance spectroscopy (EIS) measurements were conducted utilizing the aforementioned system across a frequency range from 10,000 Hz to 0.1 Hz. The Hg/HgO reference electrode was calibrated using a reversible hydrogen electrode (RHE). All test data associated with the Hg/HgO electrode were calibrated by using the Nernst equation: E_RHE_ = E_Hg/HgO_ + 0.098 + 0.0591 × pH.

### 2.3. Characterization of Material

Characterizations: The morphology was detected by using a double-beam scanning electron microscope (SEM, JEOL JEM-7800F, Tokyo, Japan) and transmission electron microscopy (TEM, JEOL JEM-2100F, Tokyo, Japan). The lattice parameters originated from X-ray diffraction (XRD, Bruker D8 ADVANCE, Billerica, MA, USA) and high-resolution TEM (HR-TEM, JEOL JEM-2100F, Tokyo, Japan). The electronic structure as well as group structure were obtained from an X-ray photoelectron spectrometer (XPS, THERMO-Fisher ESCALAB250Xi, Waltham, MA, USA) and micro-Raman spectrometer (Renishaw in Via Reflex, Dundee, IL, USA) under an excitation of 532 nm laser light. We used a LabRam HR Evolution confocal Raman microscope to perform in situ Raman spectroscopy tests to determine the active phase and dynamic surface structure of Ni_3_FeN/NF. The oxygen vacancies were tested by using an electron paramagnetic resonance spectrometer (EPR, Bryker EMXplus-6/1, Billerica, MA, USA).

## 3. Results and Discussion

### 3.1. Characterization of Composites

The XRD patterns depicted in Figure 1a corroborate the successful synthesis of Ni_3_N–CeO_2_/NF, with the observed peaks aligning well with the standard diffraction patterns of Ni_3_N (#10-0280) and CeO_2_ (#34-0394) [23,24]. The XRD patterns (Appendix A) corroborate the successful production of Ni(OH)_2_–CeO_2_/NF, with the observed peaks aligning well with the standard diffraction patterns of Ni(OH)_2_ (#14-0117) and CeO_2_ (#34-0394). Furthermore, the SEM image (Appendix A) illustrates the nanosheet morphology of the compound. Additionally, the SEM and XRD analyses displayed in Appendix A affirm that Ni_3_N/NF and CeO_2_/NF have been synthesized effectively. Figure 1b,c depict the morphology of the catalyst, whereas Figure 1d,e validate the homogeneous dispersion of the nanoparticle catalysts, each with a diameter less than 100 nm. The HRTEM images presented in Figure 1f illustrate numerous heterojunctions between Ni_3_N and CeO_2_. The lattice fringes exhibiting spacings of 0.312 nm and 0.214 nm are ascribed to the (111) plane of CeO_2_ and the (002) plane of Ni_3_N, respectively. It is observed that neighboring nanoparticles converged at the boundary seamlessly, indicating their tight interconnection, which guarantees effective electrical and mechanical interaction for stable and efficient catalysis [25,26]. The EDX spectroscopy elemental mapping of Ni_3_N-CeO_2_/NF demonstrates a uniform dispersion of nickel (Ni), cerium (Ce), oxygen (O), and nitrogen (N) throughout the analyzed area, as depicted in Figure 1g. The EDX mapping presented in Figure 1g illustrates that the Ni_3_N nanoparticles are in complete contact with CeO_2_, evidenced by the extensive overlap in the mapping. This configuration facilitates the formation of numerous heterogeneous interfaces and increases the availability of active sites, thereby enhancing catalytic activity. The EDX spectroscopy micrographs presented in Appendix A substantiate the presence of nickel (Ni), cerium (Ce), oxygen (O), and nitrogen (N) elements within the analyzed specimen.

The XPS was utilized to elucidate the surface chemical compositions of Ni_3_N-CeO_2_/NF, Ni_3_N/NF, and CeO_2_/NF. The comprehensive XPS spectra shown in Figure 2a reveal the presence of Ni, Ce, N, O, and C elements in Ni_3_N-CeO_2_/NF, whereas Ni_3_N/NF contains only Ni, O, N, and C, and CeO_2_/NF comprises Ce, O, and C. In Figure 2b, the Ce 3d XPS spectra of Ni_3_N-CeO_2_/NF reveal peak distributions spanning 883–894 eV and 896–924 eV, respectively, associated with the Ce 3d_5/2_ and Ce 3d_3/2_ orbital states [27,28]. These peaks suggest the coexistence of Ce^3+^ and Ce^4+^ states, along with satellite features. The Ce 3d_5/2_ peak in Ni_3_N-CeO_2_/NF exhibits a downward shift of approximately 0.88 eV when compared to CeO_2_/NF. The high-resolution Ni 2p spectrum of Ni_3_N-CeO_2_/NF, illustrated in Figure 2c, delineates peaks at 871.0 and 853.3 eV, attributed to Ni 2p_1/2_ and Ni 2p_3/2,_ respectively, associated with Ni-N bonding [29,30]. Compared to Ni_3_N/NF, the Ni 2p spectrum of Ni_3_N-CeO_2_/NF shows an upward shift by about 0.6 eV, suggesting electron donation from Ni^2+^ to Ce^4+^ across the heterojunction. As depicted in Figure 2d, the O1s X-ray photoelectron spectroscopy (XPS) spectra decompose into three distinct peaks: O1, O2, and O3, which correspond to metal–oxygen bonds, oxygen vacancies, and adsorbed surface oxygen, respectively [31,32]. Particularly, Ni_3_N-CeO_2_/NF exhibits a higher proportion of the O2 peak compared to CeO_2_/NF, indicating an enhancement in oxygen vacancy formation due to the presence of Ni_3_N. Based on this, we conducted EPR tests on both Ni_3_N-CeO_2_/NF and CeO_2_. The comparative results indicate that the oxygen vacancy content in Ni_3_N-CeO_2_/NF is greater than that in CeO_2_, further confirming that the presence of Ni_3_N enhances the formation of oxygen vacancies and verifying the substantial oxygen vacancies in Ni_3_N-CeO_2_/NF (Figure 2e). Furthermore, the XPS N 1s spectra of Ni_3_N-CeO_2_/NF, presented in Figure 2f, exhibit two distinct peaks at 397.7 and 399.5 eV, which correspond, respectively, to N–Ni and N–H bonding [33,34].

### 3.2. Electrocatalytic Performance for HER

To investigate the influence of the Ce/Ni molar ratio on the HER activity, a variety of samples with varying ratios (Ni_3_N-xCeO_2_/NF) were synthesized. The sample Ni_3_N-2CeO_2_/NF, where Ce/Ni = 2, demonstrates optimal HER performance, as depicted in Figure 3f. Consequently, Ni_3_N-2CeO_2_/NF was selected for subsequent experiments and denoted as Ni_3_N–CeO_2_/NF.

In contrast to others, the Ni_3_N-CeO_2_/NF composite exhibits a markedly reduced overpotential of just 42 mV at 10 mA cm^−2^ and 115 mV at 50 mA cm^−2^. This is substantially lower compared to the overpotentials recorded for the pristine Ni_3_N (156 mV and 230 mV), and CeO_2_ (186 mV and 260 mV), as depicted in Figure 3a. To assess the kinetics of the HER, the Tafel slope was calculated [35]. Figure 3b illustrates through Tafel plots that Ni_3_N-CeO_2_/NF shows the smallest Tafel gradient at 43.2 mV decade^−1^, which implies rapid mass transfer kinetics. The C_dl_, ascertained in the non-Faradaic region through cyclic voltammetry (Appendix A), was employed to assess the ECSA. The activated Ni_3_N-CeO_2_/NF exhibits the peak C_dl_ measurement of 31.7 mF cm^−2^, surpassing the levels of Ni_3_N (26.0 mF cm^−2^), and CeO_2_ (22.8 mF cm^−2^) (Figure 3c). Moreover, EIS was employed as an investigative tool to probe the reaction kinetics of the Ni_3_N-CeO_2_/NF composite, utilizing a three-electrode cell configuration in conjunction with a 1 M KOH electrolyte solution. When compared to others, Ni_3_N-CeO_2_/N exhibits a significant decrease in its charge-transfer resistance at an overpotential of 100 mV (Figure 3d). The findings from EIS indicate that Ni_3_N-CeO_2_/N demonstrates the quickest charge-transfer mechanism among the four catalysts, aligning well with its superior HER efficiency and minimal Tafel slope. The Ni_3_N-CeO_2_/NF composite exhibits robust stability, maintaining performance for 34 h at 50 mA cm^−2^ (Figure 3e). Post-stability testing, both SEM and XRD analyses (Appendix A), further confirmed its structural integrity [36].

### 3.3. Electrocatalytic Performance for OER

The sample Ni_3_N-2CeO_2_/NF, where Ce/Ni = 2, demonstrates optimal OER performance, as depicted in Figure 4f. Consequently, Ni_3_N-2CeO_2_/NF was selected for subsequent experiments and denoted as Ni_3_N–CeO_2_/NF. In contrast to others, the Ni_3_N-CeO_2_/NF composite exhibits a markedly reduced overpotential of just 350 mV at 50 mA cm^−2^. This is substantially lower compared to the overpotentials recorded for the pristine Ni_3_N (430 mV), and CeO_2_ (450 mV), as depicted in Figure 4a. The reverse scanned LSV curve is used to avoid oxidation peak effects during the OER and determine the overpotential of the electrocatalyst at a low-current density (Appendix A). To assess the kinetics of the OER, the Tafel slope was calculated [37]. Figure 4b illustrates through Tafel plots that Ni_3_N-CeO_2_/NF shows the smallest Tafel gradient at 65.2 mV decade^−1^. The electrochemical double-layer capacitances (C_dl_), ascertained in the non-Faradaic region through cyclic voltammetry (Appendix A), were employed to assess the ECSA. The activated Ni_3_N-CeO_2_/NF exhibits the peak C_dl_ measurement of 32.3 mF cm^−2^, surpassing the levels of Ni_3_N (26.3 mF cm^−2^), and CeO_2_ (21.9 mF cm^−2^) (Figure 4c). Moreover, EIS was employed as an investigative tool to probe the reaction kinetics of the Ni_3_N-CeO_2_/NF composite, utilizing a three-electrode cell configuration in conjunction with a 1 M KOH electrolyte solution. When compared to others, Ni_3_N-CeO_2_/N exhibits a significant decrease in its charge-transfer resistance at an overpotential of 300 mV (Figure 4d). The findings from EIS indicate that Ni_3_N-CeO_2_/N demonstrates the quickest charge-transfer mechanism among the four catalysts, aligning well with its superior OER efficiency and minimal Tafel slope. The Ni_3_N-CeO_2_/NF composite demonstrates substantial stability, sustaining performance for 40 h at 50 mA cm^−2^, as illustrated in Figure 4e. Subsequent stability assessments using SEM and XRD (Appendix A) further verified its enduring stability, with no detectable structural alterations [38].

### 3.4. In Situ Raman Spectroscopy

Prior studies have demonstrated that metal nitrides typically undergo a phase transformation during oxygen evolution reaction (OER) testing, thereby generating the genuine active sites of metal oxyhydroxides. In this context, we conducted in situ Raman spectroscopic analyses to investigate the structural dynamics of Ni_3_N-CeO_2_/NF post-OER testing. The in situ Raman spectra depicted in Figure 5a reveal that upon elevating the voltage to 1.34 V relative to the reversible hydrogen electrode, the peak initially located at 544 cm^−1^ progressively shifts to 553 cm^−1^, signaling the onset of oxidation in Ni_3_N, corroborating the peak observed (Figure 5a) in the linear sweep voltammetry. Upon increasing the voltage to 1.39 V, two distinct peaks are observed at 476 and 558 cm^−1^, which are attributed to the E_g_ and A_1g_ vibrational bands of NiOOH, respectively. A comparative analysis of the NiOOH vibrational peak at 558 cm^−1^ in Figure 5a,b reveals that when CeO_2_ is present, the half-peak width of NiOOH increases by approximately 20%. This suggests that CeO_2_ promotes the development of NiOOH that is enriched with a higher density of oxygen vacancies.

### 3.5. Overall Water Splitting in an Alkaline Electrolyte

Given the superior electrochemical properties of the HER and OER, Ni_3_N-CeO_2_/NF, once activated, acts as dual-function catalysts, facilitating the formation of a two-electrode system for comprehensive water division (Figure 6a) [39,40]. At 10 mA cm^−2^, Ni_3_N-CeO_2_/NF||Ni_3_N-CeO_2_/NF demonstrates a low cell voltage of 1.52 V in 1 M KOH at 25 °C (Figure 6a). It is observed that Ni_3_N-CeO_2_/NF’s efficiency surpasses that of Pt/C/NF||IrO_2_/NF (1.63 V) and other previously documented electrocatalysts, underscoring the feasibility of effective overall water splitting (Figure 6c) [41,42,43]. Furthermore, the battery has demonstrated remarkable steadiness. At 10 mA cm^−2^, the battery can sustain its current density for over 34 h without experiencing voltage loss, and it retains 93.52% of it (Figure 6b).

## 4. Conclusions

In conclusion, we have developed bifunctional heterojunction catalysts containing abundant oxygen vacancies. The material demonstrates superior catalytic proficiency in the context of water electrolysis, achieving remarkably low overpotentials of merely 42 mV at 10 mA cm^−2^ for the HER. Particularly, the Ni_3_N-CeO_2_/NF exhibits an ultra-low overpotential for the OER (η_50_ = 350 mV). Through EPR testing, we can see that the catalyst also has abundant oxygen vacancies. The in situ Raman analysis distinctly demonstrated that cerium dioxide (CeO_2_) significantly enhanced the formation of nickel oxyhydroxide (NiOOH) with a high concentration of oxygen vacancies (O_v_). This study furnishes innovative perspectives that may inform the future conceptualization of electrocatalysts characterized by enhanced activity, stability, and bifunctionality for the application of water electrolysis.

## Data Availability

The data are contained within the article and Appendix A.

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
