# Peer review of "Oxygen Vacancy-Enhanced Ni3N-CeO2/NF Nanoparticle Catalysts for Efficient and Stable Electrolytic Water Splitting"

_nanomaterials, 2024, doi:10.3390/nano14110935_

Round 1

Reviewer 1 Report

Comments and Suggestions for Authors

In this work the authors synthetize  Ni3N-CeO2/Nf heterostructures and analyse the corresponding performance for both Oxygen and Hydrogen evolution reaction.

Here are my comments:

1.       Abstract: Is it possible to add reference in the abstract? the same holds for conclusions.

2.       Figure 1 please remove the red underline from each figure in the word nm.

3.       How is the sample prepared for TEM observation?

4.       Figure 3: How many ohms is the Resistance adopted for the iR correction? May I see the uncorrected curves? Thank you

5.       Is it possible to have same colour and same marker for the same sample in the figure? It is difficult for me to evaluate the comparison.

6.       Figure 3: Is it possible to enlarge the ylim of the stability plot? (for example from -0.5V to 0.5V).

7.       The same of 4 and 5 but for figure 4.

Comments on the Quality of English Language

English language is correct for me

Reviewer 2 Report

Comments and Suggestions for Authors

In this manuscript, the authors reported a facile hydrothermal technique followed by a subsequent nitridation process to prepare a Ni3N-CeO2/NF heterostructure as a binfunctional electro-catalyst toward the OER and HER for water splitting. By using in situ Raman spectroscopic analysis, it was found that CeO2 facilitates the rapid reconfiguration of oxygen vacancy-enriched nickel oxyhydroxide (NiOOH), thereby enhancing the OER performance. These results can have implications for development of highly efficient and cost-effective electro-catalysts for water electrolysis. Overall, this work has good novelty and is worthy to be published in Nanomaterials. To further improve the quality of the manuscript, the below detailed comments need to be addressed.

1. It is not common for the journal’s Abstract to include references. Please double check authors guidelines to make revisions if necessary.

2. “Water-splitting” (with a hyphen) is often used as a descriptive adjective, hence not appropriate for inclusion as a keyword. “Water splitting” is more suitable in this case.

3. To appeal to a broader readership, related works on water electrolysis can be included in Introduction (e.g., InfoMat, 2024, DOI: 10.1002/inf2.12494; Materials Reports Energy, 2022, 2, 100144).

4. Figure 3a-d, because the same set of samples are studied for these figures, it is suggested that the authors apply the same sample labelling and coloring to ensure consistency and avoid misunderstanding. The same applies to Figure 4b, whose coloring and naming are inconsistent with Figure 4a, 4c, and 4d. Also, the y axis “Overpotential” should be revised into “Potential” for Figure 4b.

5. Forming a composite structure is indeed helpful to electrocatalysis. This point can be highlighted in the Introduction. Related works are recommended to be cited (e.g., Small, 2021, 17, 2101573).

6. Scheme 1 needs some minor revision. In the description of the reaction conditions, the punctuation between temperature and time should be revised into comma.

7. How was the potential vs mercury/mercuric oxide electrode converted to that vs the RHE? This information should be provided in the Experimental section.

8. Figure 1c-g, there is a red curve near the scale bars. Please correct.

9. Figure 6 discusses the overall water splitting performance in a two-electrode setup, then the “potential” term in these figures is better to be revised into “Cell Voltage”.

Reviewer 3 Report

Comments and Suggestions for Authors

In this work, the authors prepared oxygen-vacancy-rich Ni3N-CeO2/NF nanocomposites for efficient and stable alkaline water splitting. Various characterizations including in-situ Raman were applied to demonstrate the performance origin. However, some important issues must be solved before acceptance.

1.     The full names of abbreviations should be given when they first appear. The spacing should be added between values and units. Please check the style of references and no references should be added in the abstract. The related reference of XRD PDF cards should be provided. In Figure 1a, the XRD pattern of NF should be presented.

2.     In the introduction part, brief comparisons between TMNs and other P, S, Se-containing compounds should be given to demonstrate the advantages of TMNs, please refer to 10.1016/j.cej.2023.141674 for this point.

3.     For hybrid materials, the detailed content of each phase should be identified via XRD refinement or mapping.

4.     Please explain why Ce4+ locates at lower energy position as compared with Ce3+. No references are provided for the O 1s XPS analysis. Please refer to 10.1039/D1TA10652J for this point.

5.     The counterpart samples were not measured using EPR as comparisons. To exclude the possible oxidation current or pseudocapacitance effect from LSV, the authors are suggested to perform LSV using negative scans.

Comments on the Quality of English Language

Minor editing of English language required

Reviewer 4 Report

Comments and Suggestions for Authors

The article successfully synthesizes a Ni3N-CeO2/NF heterostructure with abundant oxygen vacancies, contributing to its enhanced catalytic activity. The catalysts demonstrated low overpotentials and significant stability, making them suitable for water-splitting applications. The comprehensive in situ Raman spectroscopy analysis adds depth to understanding the catalytic mechanisms at play.

Points of Improvement:

1.       The disparity in the Cdl values for the same electrodes in Figures 3c and 4c raises concerns. If these measurements are within a range with no Faradaic reactions, the Cdl values should be consistent. The variation suggests potential issues in the experimental setup or measurement errors that need to be addressed. The reported values are also quite large, indicating an ECSA greater than the 1 cm² used for presenting the results. Clarifying the actual surface area used and possibly re-evaluating the measurements would be beneficial.

2.       The scales for potential in Figures 3d, 4d, and 6b are too broad, making it difficult to observe the fine details of the results. Using smaller ranges would provide a clearer view of the data, allowing for a more precise interpretation of the electrochemical behaviour and performance of the catalysts.

Round 2

Reviewer 1 Report

Comments and Suggestions for Authors

I think that the revised manuscript should be accepted for publication on Nanomaterials.

The authors have properly addressed to all my issues.